# Superconducting Gravimeters: A Novel Tool for Validating Remote Sensing Evapotranspiration Products

Jonatan Pendiuk [1,2], María Florencia Degano [2,3,*], Luis Guarracino [1,2,4] and Raúl Eduardo Rivas [3,5]

1　Facultad de Ciencias Astronómicas y Geofísicas, Universidad Nacional de La Plata, La Plata B1900, Argentina; jpendiuk@fcaglp.unlp.edu.ar (J.P.); luisg@fcaglp.unlp.edu.ar (L.G.)
2　Consejo Nacional de Investigaciones Científicas y Técnicas, Ciudad Autónoma de Buenos Aires C1425FQB, Argentina
3　Instituto de Hidrología de Llanuras "Dr. Eduardo Jorge Usunoff", Tandil B7000, Argentina; rrivas@rec.unicen.edu.ar
4　Facultad de Ciencias Naturales y Museo, Universidad Nacional de La Plata, La Plata B1900, Argentina
5　Comisión de Investigaciones Científicas de la Provincia de Buenos Aires, La Plata B1900, Argentina
*　Correspondence: mfdegano@ihlla.org.ar; Tel.: +54-924-9438-5520

**Abstract:** The practical utility of remote sensing techniques depends on their validation with ground-truth data. Validation requires similar spatial-temporal scales for ground measurements and remote sensing resolution. Evapotranspiration (ET) estimates are commonly compared to weighing lysimeter data, which provide accurate but localized measurements. To address this limitation, we propose the use of superconducting gravimeters (SGs) to obtain ground-truth ET data at larger spatial scales. SGs measure gravity acceleration with high resolution (tenths of nm s$^{-2}$) within a few hundred meters. Similar to lysimeters, gravimeters provide direct estimates of water mass changes to determine ET without disturbing the soil. To demonstrate the practical applicability of SG data, we conducted a case study in Buenos Aires Province, Argentina (Lat: $-34.87$, Lon: $-58.14$). We estimated cumulative ET values for 8-day and monthly intervals using gravity and precipitation data from the study site. Comparing these values with Moderate Resolution Imaging Spectroradiometer (MODIS)-based ET products (MOD16A2), we found a very good agreement at the monthly scale, with an RMSE of 32.6 mm month$^{-1}$ (1.1 mm day$^{-1}$). This study represents a step forward in the use of SGs for hydrogeological applications. The future development of lighter and smaller gravimeters is expected to further expand their use.

**Keywords:** remote sensing; ground-truth data; validation; superconducting gravimeters; evapotranspiration





## 1. Introduction

Evapotranspiration is one of the major components of the global water cycle and provides a critical link between terrestrial water, carbon, and surface energy exchanges. Accurate estimation of ET is essential for agrometeorological studies, understanding hydrological and ecological processes, and developing effective water management strategies [1–3]. However, ET is inherently difficult to measure and predict especially at large spatial scales as its spatio-temporal patterns depend on vegetation types, soil properties, and the meteorological conditions at the study site [4].

Most methods for estimating ET are based on the water balance equation. This equation, also known as the continuity equation, accounts for changes in water storage in a given water system and time interval caused by inputs and outputs of water flow (i.e., rainfall, ET, and surface and subsurface runoff). Water storage is probably the most difficult component of the water balance equation to quantify for two main reasons. First, water storage is composed of different contributions such as soil moisture, groundwater, and surface water bodies (including snow and ice) that are usually monitored individually [5–7].

Secondly, the high spatial heterogeneity observed in both the hydraulic properties and the scales of the flow processes involved, makes it difficult to obtain representative values of water storage changes for the whole system [8]. Thus, the estimation of ET from the water balance equation is largely conditioned by the accuracy in determining water storage changes.

At the point scale, weighable lysimeters provide accurate measurements of water storage changes in the unsaturated zone by recording mass changes of a soil profile [9]. Based on these measurements and an independent record of the flux across the lower boundary of the lysimeter, ET can be easily calculated from the mass balance equation [10]. It is important to note that weighable lysimeters are considered to be the only device that directly measure the actual ET [11,12]. However, lysimeter data describe the ET at a single point in the field and do not account for water storage changes in deeper zones of the soil [13].

Water storage changes can also be estimated at regional and global scales using data from the Gravity Recovery and Climate Experiment (GRACE) and GRACE Follow-On (GRACE-FO) satellite missions [14]. These missions provide high-precision, time-variable measurements of the Earth's gravity field that have been used in many studies of mass variation in different components of the climate system, including terrestrial water storage changes, groundwater depletion, ice sheet and glacier changes, and ET [15–18]. Unfortunately, the application of GRACE data to local water resources is severely limited by its the coarse spatial resolution. Swenson et al. [19] showed that the accuracy of GRACE-derived water storage increases with the study area and they estimated an accuracy of less than 1 cm for study areas larger than 400,000 km$^2$. Based on this result, the spatial resolution of GRACE data is not expected to be better than a few hundred km.

Despite the above-mentioned methods, the determination of water storage changes at the mesoscale (tens to hundreds of meters) remains a challenge for the hydrological community. Measurements of water system components at this spatial resolution are useful for validating remote sensing estimates [20–22]. A promising method for estimating water storage at the mesoscale is to use data from superconducting gravimeters (SGs) [8,23]. Similar to the GRACE and GRACE-FO satellite missions, these terrestrial instruments measure temporal variations of gravity with unprecedented accuracy and a spatial resolution estimated to be between 50 and 4000 m [23]. From these gravity measurements, it is possible to estimate the integrated water storage changes with an accuracy on the order of a few mm. Due to these characteristics, SGs have been used as hydrological sensors for calibration and validation of hydrological models [24], for monitoring the artificial recharge of an aquifer [25], and for estimating specific yields [26]. In addition, SG data can be used to estimate ET at the mesoscale [8,27–30]. Van Camp et al. [27] applied a stacking process to gravity time series for rain-free periods to isolate the effect of ET on the gravimetric signal and thus estimate the daily mean ET rate. Güntner et al. [8] proposed an inversion process based on the water balance equation to estimate ET at annual and daily scales using potential evapotranspiration values as input data. These ET estimates were then validated with data recorded by a lysimeter. Carrière et al. [28] estimated daily ET in a rain-free period using data from two superconducting gravimeters vertically separated by a distance of 512 m. This arrangement between the gravimeters eliminates the non-hydrological large-scale effects observed by both instruments, improving the analysis of local water storage changes.

Over time, several strategies have been proposed for validating actual ET products obtained from satellite data. In general, these validations are carried out using field data from weighing lysimeters [30–33], as well as flux tower data [34–36], which are the main and most widely distributed databases of local measurements. Remote sensing methods represent a valuable tool for ET monitoring as they allow us to map the patterns of this hydrological variable at different spatial scales. In particular, the MOD16A2 global terrestrial ET product from the Moderate Resolution Imaging Spectroradiometer (MODIS) is widely used in different ecosystems due to its high spatio-temporal resolution [37]. The

practical utility of the MOD16A2 product depends on the validation with real ground-based data at similar temporal and spatial scales [38–41]. Therefore, the main objective of this study is to demonstrate the potential of SGs as a novel tool for validating estimates of the MOD16A2 actual ET product on different temporal scales. Ground-truth ET was obtained from the water balance equation using SG data recorded at the Argentine–German Geodetic Observatory (AGGO) located in the Buenos Aires Province, Argentina, for an annual period from May 2017 to May 2018. The Argentine Pampas region, which is predominantly flat and sub-humid, is an ideal study site for assessing the MOD16A2 product using accurate estimates that include spatial variations in coverage (e.g., soil, crops, and trees).

## 2. Study Site and Methodology

### 2.1. Study Site

AGGO is a fundamental geodetic observatory located in Parque Pereyra Iraola, Berazategui, Argentina (Figure 1). AGGO is equipped with the main geodetic techniques such as VLBI, SRL, GNSS, absolute gravimetry, and a superconducting gravimeter (SG038) which has been continuously measuring temporal gravity changes since December 2015 [42]. It is worth noting that, the SG038 is currently the only SG operating in Latin America and the Caribbean. In addition to geodetic techniques, hydrometeorological sensors have been installed at AGGO. A weather station with sensors for air temperature, relative humidity, short and long wave radiation, wind speed, and two rain gauges were installed in April 2016. Soil moisture sensors were installed in several profiles between 5 and 450 cm depth and are located in the vicinity of the SG038. The hydrometeorological dataset is available in Mikolaj et al. [43].

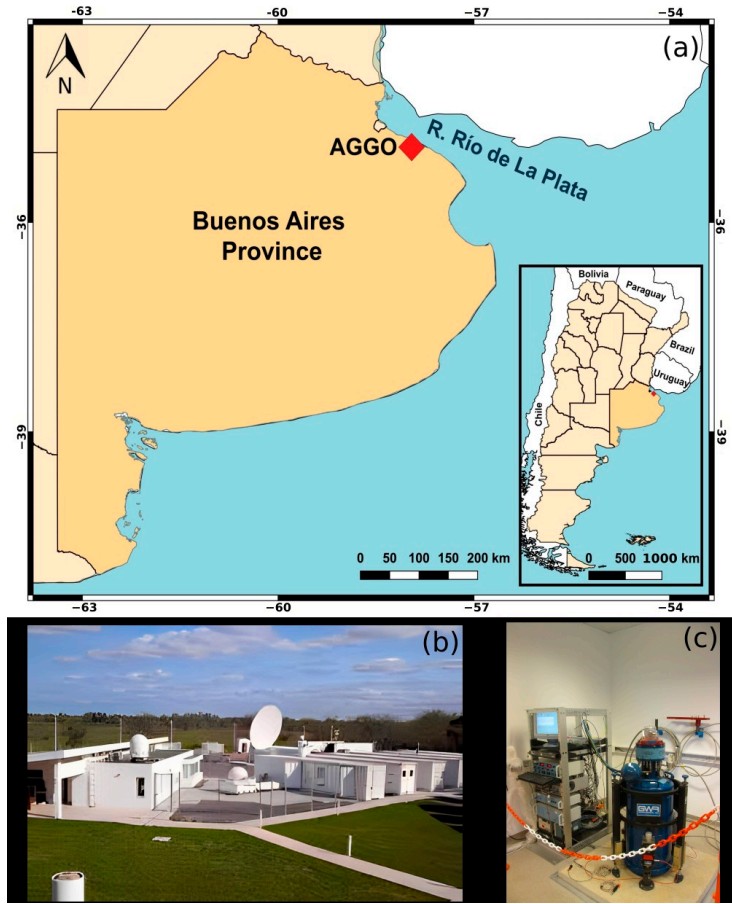

**Figure 1.** (**a**) Location of AGGO in Buenos Aires Province, Argentina, (**b**) panoramic view of AGGO, and (**c**) superconducting gravimeter SG038.

AGGO is located in the Pampas region, one of the largest plains in the world (approximately 520,000 km$^2$), which is an important food-producing region due to its fertile land that is suitable for livestock and agricultural production [44]. In this region, vertical water flow predominates over horizontal flow with a strong interaction between surface water and groundwater. Furthermore, ET is a key hydrological parameter at the study site as it is the main mechanism of water flow and it is strongly related to with water availability.

The AGGO site is covered with natural grass and surrounded by a eucalyptus forest. The climate is temperate, with a mean annual temperature, relative humidity, and rainfall of 15.8 °C, 74.3%, and 1007 mm, respectively; the soil is composed of silty and clayey sediments and the mean groundwater level is about 13 m deep.

### 2.2. Superconducting Gravimeter

SGs are the most precise and long-term stable relative instrument for monitoring temporal gravity variations. The measurement resolution of SGs is in the range of 0.1–0.3 nm s$^{-2}$ with a time resolution of 1 s. The measurement principle is based on the levitation of a superconducting sphere in a stable magnetic field generated by a pair of niobium coils [45]. In order to maintain the superconducting state, the gravity sensor is inside a liquid helium Dewar with a refrigeration system that keeps the temperature close to absolute zero. Any change in gravity produces an external force that affects the levitation of the sphere. To compensate for this external force and to keep the sphere in a fixed position, an electric current is injected through an auxiliary feedback coil inducing a change in the magnetic field intensity and consequently, in the voltage. The SG output signal is provided in voltage units, and then, it is converted into gravity values using a calibration factor.

According to Newton's law, the gravimetric signal decreases with the square of the distance. Thus, SGs are more sensitive to nearby mass variations. To define the radius of influence $R$ of an SG, we will consider the gravimetric effect of an infinite horizontal layer of water at depth z. The gravity response of a 1 mm thick layer water is 0.42 nm s$^{-2}$, which is usually referred to as *water admittance* [46] and is independent of the depth $z$ [47]. Since the SG accuracy is in the range 0.1–0.3 nm s$^{-2}$, these devices can detect water layers with thicknesses from 0.2 to 0.7 mm thick.

The radius of influence or footprint of an SG can be inferred from the comparison between the water admittance value and the gravity effect of a vertical cylinder of 1 mm thickness and radius $R$ at a depth z below the SG, as shown in Figure 2.

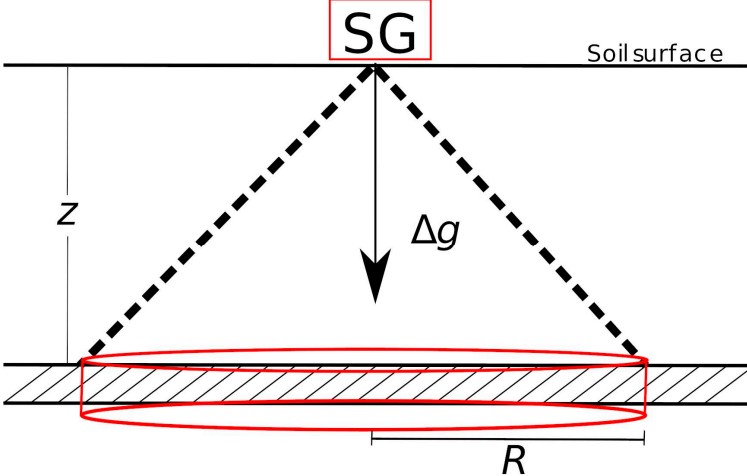

**Figure 2.** Gravimetric modelling of water storage changes at depth $z$ below an SG using an infinite horizontal layer and a vertical cylinder of radius $R$.

The gravity effect ($\Delta g$) of a vertical cylinder of radius $R$ and 1 mm thickness is estimated from the following equation [48]:

$$\Delta g(z, R) = \left(1 - \frac{z}{\sqrt{z^2 + R^2}}\right) \; 0.42 \text{ nm s}^{-2} \qquad (1)$$

Figure 3 shows the gravity effect ($\Delta g$) as a function of $R$ for different depths. Note that when $R$ tends to infinity in Equation (1), we obtain the value of water admittance (0.42 nm s$^{-2}$). However, 99% of the gravity signal comes from an area of radius 200 to 2000 m, depending on the depth. Moreover, from Equation (1) it can be shown that approximately 99% of the signal originates within a radius:

$$R = 100 \, z \qquad (2)$$

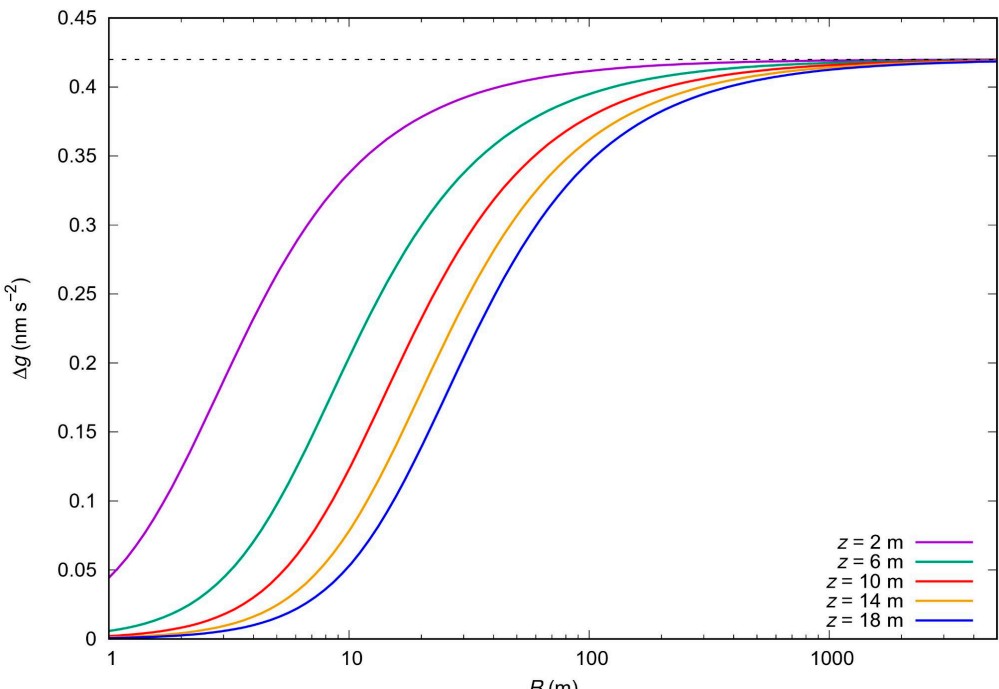

**Figure 3.** Gravity effect ($\Delta g$) of a vertical cylinder with a thickness of 1 mm and variable radii at different depths (2 to 18 m). The dashed line indicates the gravity effect of an infinite horizontal layer of water of 1 mm thickness (water admittance).

Equation (2) defines the radius of influence of the SG, which depends on the depth $z$. The dependence on depth explains why different radii of influence ranging from 50 to 4000 m are reported in the literature (e.g., [8,12,49]). Since the mean groundwater depth in the study site is 13 m, a radius of influence of less than 1300 m would be expected.

It is important to note that Equation (1) is valid for flat terrain. If the topography is not flat, specific site effects on gravity measurements must be considered to calculate $R$.

### 2.3. Ground-Truth ET Estimates Using SG Data

An ET water-balance-based approach is commonly used as a benchmark for evaluating other ET products. This approach is based on the principle of mass conservation and considers the difference between incoming and outgoing hydrological fluxes at different temporal and spatial scales. The classical water balance equation can be expressed as:

$$ET = P - R_s - \Delta S \qquad (3)$$

where ET, P, and R$_s$ are cumulative ET, precipitation, and runoff (in mm), respectively, during an arbitrary period of time $\Delta t = t_i - t_{i-1}$, and $\Delta S$ represents the water storage changes. In this study, $\Delta t$ periods of 8 days, which is the temporal resolution of the MOD16A2 product, and 1 month are considered. The estimated ET values using Equation (3) are representative within an area defined by the SG influence radius whose value is approximately 1300 m in the study site (see Section 2.2).

As previously mentioned, the study site is characterized by very low topographic slopes. Then, the runoff ($R_s$) can be estimated as:

$$R_s = \alpha P \tag{4}$$

where $\alpha$ is a model parameter which is assumed to be constant. Several studies performed in Buenos Aires Province report a value of $\alpha = 0.05$.

Water storage changes $\Delta S$ are estimated from SG data ($\Delta g$) using a factor $C$ to convert gravity variations into equivalent water storage changes as follows:

$$\Delta S = C \Delta g \tag{5}$$

The value of $C$ depends on the topography and the size of the building housing the SG. The building obstructs the processes of ET and natural infiltration, affecting the gravity measurements. This phenomenon is known as the umbrella effect [50]. If the topography of the study site is flat and the umbrella effect is negligible, the conversion factor $C$ is 2.38 mm $(\text{nm s}^{-2})^{-1}$ [46]. At the AGGO study site, Pendiuk [29] estimated a site-specific $C$ of 2.57 mm $(\text{nm s}^{-2})^{-1}$ from a regression analysis between gravity residual and water storage changes calculated from soil moisture data.

By combining Equations (3)–(5) an expression for computing ET is obtained:

$$ET = (1 - \alpha)P - C\Delta g \tag{6}$$

Note that Equation (6) depends only on precipitation and gravity residuals so no additional data about the physiological properties of the canopy are required.

The proposed method to obtain ET from gravimetric data can be summarised in the following steps:

1. Convert the raw SG data to gravity variations using the instrument-specific calibration factor.
2. Removal of non-hydrological signals such as Earth tides, atmospheric and oceanic loading, polar motion effects, and instrumental drift using different models.
3. The data processing described in the previous steps was performed by the International Geodynamics and Earth Tide Service (IGETS). This product (level 3) can be downloaded from the database of the Information System and Data Centre of the GFZ (GeoForschungsZentrum, Germany) [43].
4. Sample precipitation data and gravimetric residuals on the same timescale (e.g., 8 days, one month, etc.).
5. Calculate ET using Equation (6).

In Section 3, the ET values estimated using the methodology described above are used as ground-truth data to validate the global terrestrial ET products MOD16A2.

### 2.4. Description of the Data Set

In this work, an annual period from 1 May 2017 to 30 April 2018 was considered. The gravity residual and precipitation time series are available from the GFZ database [43]. The MODIS global terrestrial ET data (MOD16A2) were obtained from the Google Earth Engine platform.

### 2.4.1. Gravity Residual and Water Storage Change Time Series

Gravity residuals ($\Delta g$), with an hourly time resolution, are provided by the GFZ database [43]. These residuals were obtained from the SG038 data after subtraction of the Earth tides, atmospheric and oceanic loading, polar motion effects, instrumental drift, and both global atmospheric and hydrological effects [51]. Additionally, a moving average filter with a window length of 6 days was applied to the data series in order to reduce the non-hydrological effects that can be seen in the gravity residuals. Figure 4 shows gravity residual time series which is assumed to reflect the main local water storage changes in the surroundings of SG038. For the period November 2017–March 2018, the water storage is assumed to decrease significantly since the gravity residual was directly proportional to the water storage changes (see Equation (5)).

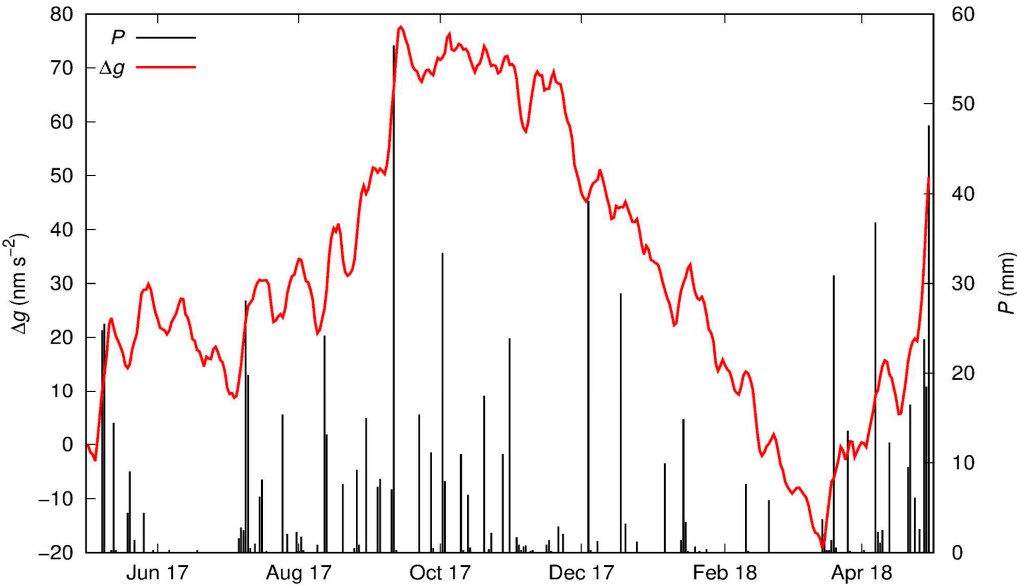

**Figure 4.** Gravity residual ($\Delta g$) and precipitation ($P$) time series.

Note that, despite the filtering applied to the gravity data, a non-hydrological signal with a period of approximately 14 days remained, identified from a spectral analysis.

### 2.4.2. Precipitation

Precipitation ($P$) is recorded at hourly time steps by two rain gauges. They are located at a distance of 10.9 m from each other near the SG038. Both time series are pre-processed and then merged to obtain a precipitation series without missing values [51]. Figure 4 shows the precipitation series for the study period. The cumulative precipitation was 838 mm, which is less than the mean annual value (1007 mm). April 2018 was the rainiest month of the study period ($P$ = 180 mm), followed by September and October 2017, while June was the driest month ($P$ = 1.1 mm).

### 2.4.3. Satellite Data (MOD16A2)

MOD16 was selected for this study since, to our knowledge, it is the most widely used remote-sensing-based ET data set at the global scale. MOD16 is usually used to evaluate modelled ET due to its high spatial resolution, extensive historical data availability, and global coverage. However, it is not applicable in ice-covered catchments, desert areas, and water bodies where the MOD16 algorithm cannot derive ET [52]. The MOD16 product has been evaluated against ground measurements derived from eddy flux towers in several parts of the world, resulting in different levels of accuracy in different regions, including North America, Europe, South America, and Asia [52–55]. In all of these compar-

isons, it is important to highlight that the ground measurements themselves have inherent uncertainties, typically ranging from 10% to 30% when ET is estimated by eddy flux towers.

The MOD16A2 version 6 data were provided by Earth Data from the National Aeronautics and Space Administration (NASA). The algorithm used to calculate ET is based on the Penman–Monteith equation [56]. The inputs come from the reanalysis of the global daily meteorological dataset Modern-Era Retrospective analysis for Research and Applications (MERRA), and from MODIS data: a land cover (MOD12Q1) product, the Leaf Area Index/Fraction of Photosynthetically Active Radiation (LAI/FPAR MOD15) product, and an albedo (MCD43A2/A3) product. MOD16A2 is based on the algorithm first proposed by Mu et al. [46]. Actual ET is retrieved from remote sensing data as the sum of evaporation losses from the wet canopy, actual plant transpiration, and actual soil evaporation [52,57]. The MOD16A2 estimates ET at 8-day time intervals. This time period captures changes in ET patterns on a global scale. Additionally, the 8-day interval is also related to the satellite data acquisition used in the product, resulting in good accuracy. However, some updates have been implemented in the operational code to fix some issues. These updates are detailed in Running et al. [58].

The MOD16A2 version 6 product has a spatial resolution of 500 m per pixel. To obtain representative ET values for the study site, average values were computed for a $3 \times 3$ pixel grid with the SG as the central point. Note that the grid size of $3 \times 3$ pixel (1500 m) is similar to the radius of influence of the SG, which is approximately 1300 m (see Equation (2)).

*2.5. Statistical Performance Metrics*

To compare the performance of the MOD16A2 products in relation to the ET estimated by the gravity-based approach, three statistical metrics were computed: (1) Mean Absolute Error (MAE), (2) Root Mean Square Error (RMSE), and (3) Pearson correlation coefficient (r):

$$MAE = \sum_{j=1}^{N} \frac{\left| ET_M^j - ET_{SG}^j \right|}{N} \tag{7}$$

$$RMSE = \sqrt{\sum_{j=1}^{N} \frac{\left( ET_M^j - ET_{SG}^j \right)^2}{N}} \tag{8}$$

$$r = \frac{\sum_{j=1}^{N} \left( ET_M^j - <ET_M> \right) \left( ET_{SG}^j - <ET> \right)}{\sqrt{\sum_{j=1}^{N} \left( ET_M^j - <ET_M> \right)^2 \left( ET_{SG}^j - <ET_{SG}> \right)^2}} \tag{9}$$

where N represents the number of total observations, and $ET_M^j$ and $ET_{SG}^j$ are the MOD16A2 data and ET gravity-based approach at time period j, respectively. The symbol <> indicates the mean value of the variable for the whole period of the analysis.

RMSE and MAE are two of the most commonly used metrics in model evaluation in the physical and environmental sciences [59,60]. These statistical metrics are easy to interpret, as lower values indicate better model performance. One of the main differences between these metrics is that the RMSE is more sensitive to outliers and extreme values. However, both metrics should be used together since there is no consensus on which metric is the most appropriate for understanding the agreement between ET estimates. The Pearson correlation coefficient r quantifies the linear correlation between $ET_M$ and the ET gravity-based approach.

### 3. Results

*3.1. ET Estimates on an 8-Day Timescale*

ET was estimated by the gravity-based method using Equation (6) on an 8-day timescale and compared to the MOD16A2 product ($ET_M$). Figure 5 shows the ET gravity-based approach ($ET_{SG}$) and $ET_M$ time series. Negative values of $ET_{SG}$ are not realistic

and were removed from this analysis. $ET_M$ showed a clear seasonal pattern where the maximum values were observed in the period November 2017–December 2018, whereas the values of $ET_{SG}$ showed dispersion. Moreover, the gravity-based technique yielded high ET values for the cold period May 2017–June 2017. The statistical metrics were calculated using Equations (7)–(9) and the obtained values are listed in Table 1. The estimated MAE and RMSE values were 9.32 (1.2 mm day$^{-1}$) and 11.9 mm (1.5 mm day$^{-1}$), respectively. In addition, the estimated value of the correlation coefficient was $r = 0.4$, indicating a weak correlation due to the significant dispersion of $ET_{SG}$.

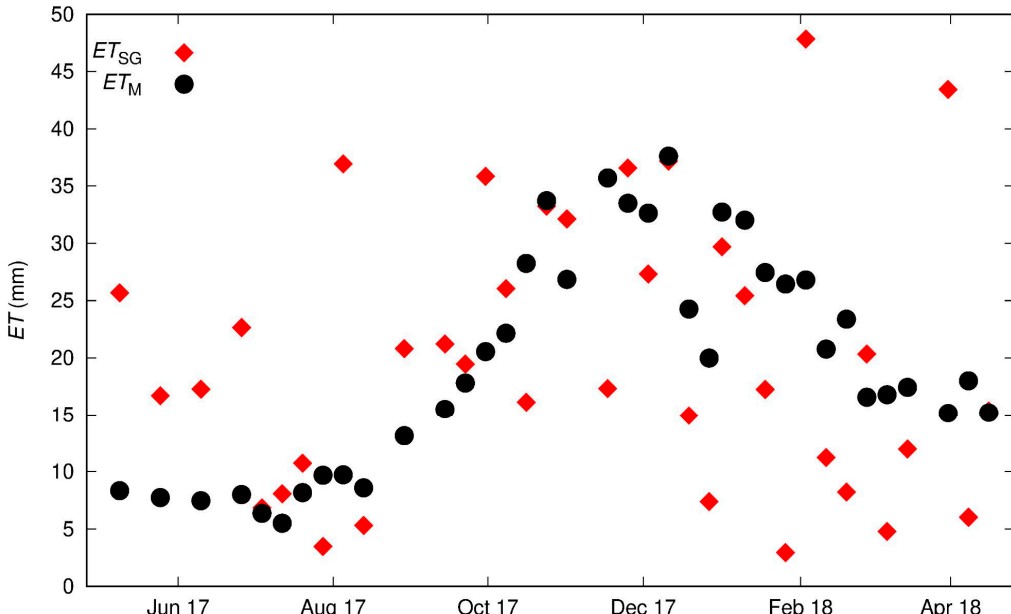

**Figure 5.** ET gravity-based approach ($ET_{SG}$) and MOD16A2 ($ET_M$) at an 8-day-scale time series.

**Table 1.** Estimated statistical metrics at 8-day and monthly scales.

| Coefficient (Units) | Value at 8-Day Scale | Value at Monthly Scale |
|---|---|---|
| MAE (mm) | 9.32 | 24.5 |
| RMSE (mm) | 11.9 | 32.6 |
| r (-) | 0.4 | 0.7 |

In order to visualise the dispersion between the ET series, a crossplot of $ET_{SG}$ and $ET_M$ is shown in Figure 6. The discrepancies between the ET series could be attributed to a non-hydrological signal with a period of approximately 14 days that was observed in the gravity data (Figure 4). This periodical signal affecting the gravimetric residuals may be associated with the global non-tidal oceanic and atmospheric loading effect or spring tides, which were probably not completely removed in the gravity signal processing stage. At timescales longer than 14 days, this signal is expected to attenuate. In the next section, ET estimated by both methods at a monthly timescale are compared.

### 3.2. Estimation of Monthly ET

In this section, $ET_{SG}$ was estimated from Equation (6) using cumulative precipitation and gravity residual data with a monthly time period $\Delta t$. To obtain MOD16A2 ET values on a monthly timescale, a temporal upscaling approach was applied to the 8-day $ET_M$. Figure 7 shows monthly values of $ET_{SG}$ and $ET_M$ and now no dispersion was observed. The ET estimated by both methods showed a seasonal pattern where minimum and maximum values were observed in winter and summer, respectively. Note that ET estimates were similar for the winter season. The largest discrepancies between ET gravity-based approach

and $ET_M$ were observed in the spring–summer period, with values ranging from 42 to 59 mm. In addition, $ET_M$ values tended to be higher than $ET_{SG}$ values. This period coincided with the beginning of a marked drop in water storage observed by the soil moisture sensors installed in AGGO and which can also be inferred from the gravimetric residuals shown in Figure 4.

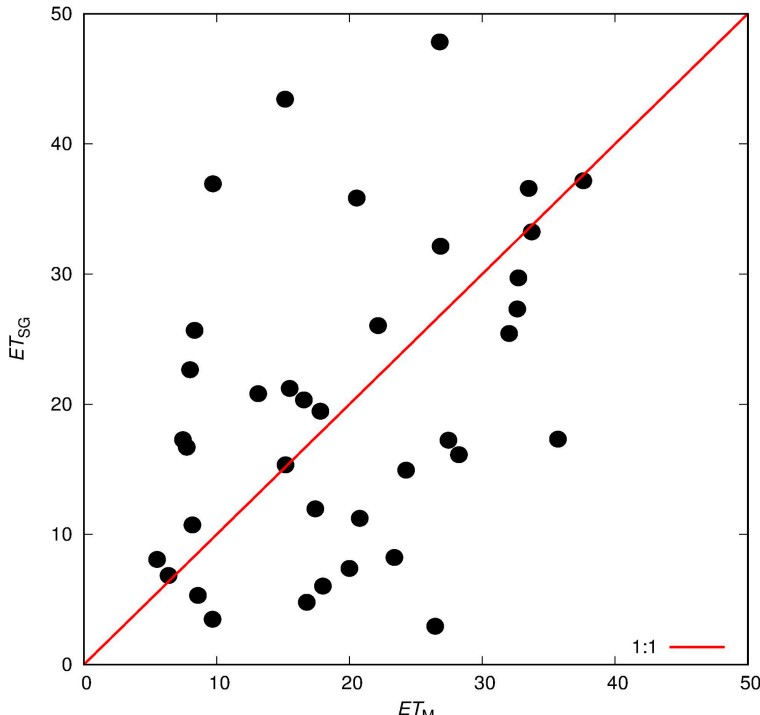

**Figure 6.** Crossplot between MOD16A2 ET ($ET_M$) and ET gravity-based approach ($ET_{SG}$) at 8-day timescale.

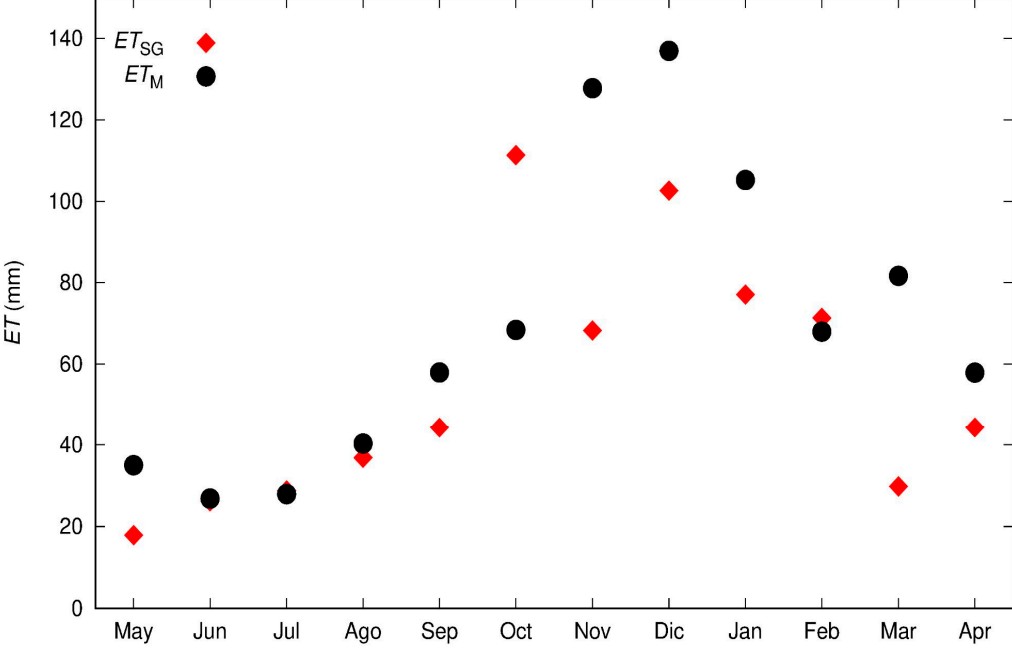

**Figure 7.** Comparison between ET gravity-based approach and MOD16A2 at monthly scale.

At the monthly timescale, the values of all statistical metrics improved. Certainly, MAE (24.5 mm = 0.8 mm day$^{-1}$) and RMSE (32.6 mm = 1.1 mm day$^{-1}$) values were reduced compared to those obtained at the 8-day timescale (see Table 1). In addition, a higher value of the Pearson correlation coefficient $r = 0.7$ was obtained, indicating a positive monotonic relationship between ET$_M$ and ET$_{SG}$.

## 4. Discussion

The examples in the previous section show the potential of SGs to provide ground-truth ET data as well as the current limitations in its practical implementation. To the best of our knowledge, these results represent the first attempt to validate satellite ET products with SG data.

One of the main advantages of the proposed validation is the similarity of the spatial scales of both techniques, which is estimated to be hundreds of metres. In Section 2.2, a relationship was proposed to estimate the radius of influence of the SG, which can be approximated by one hundred times the water table depth.

However, the validation results show some discrepancies on different timescales. For short time periods (8 days), a large dispersion is observed in the ET data obtained with the SG. This dispersion was mainly attributed to a 14-day non-hydrological component that persists in the gravimetric data. The period of this signal can be associated with the spring tides caused by the gravimetric attraction of the Moon [61]. This undesired signal highlights the need to improve gravimetric data processing to remove all non-hydrological components. Fortunately, the validation results improved significantly on a monthly scale. This improvement was due to the fact that the effect of the 14-day component was attenuated when the values are integrated over a 30-day period.

To verify the effect of the timescale on ET gravity-based approach estimates, an additional comparison at 16 days was also performed. The comparison of both data sets gave values of MAE = 13.5 mm (0.8 mm day$^{-1}$), RMSE = 19.6 mm (1.2 mm day$^{-1}$), and $r = 0.58$. Note that these statistical values improved compared to those obtained for the 8-day timescale, but are worse than those of the monthly scale (see Table 1). These results confirm that a better agreement between the ET series is observed as the timescale increases.

The main differences between both ET monthly estimates can be attributed to the MOD16A2 algorithm. Specifically, the MOD16A2 algorithm does not directly account for soil moisture variations when estimating ET. Instead, it is constrained by the daily vapour pressure deficit, relative humidity, and minimum air temperature [46]. This approach is based on the assumption of congruence between near surface atmosphere and soil moisture conditions at coarser spatio-temporal scales [62,63]. The SG method does not require any constraints related to soil moisture to estimate ET. SGs implicitly record the variations in soil moisture, so there is no need to measure or model the water content in the soil profile.

To date, SG data have been used mainly for geodetic and geodynamic purposes. The use of SGs as a hydrogeological tool is relatively recent but has great potential for future work. SGs are the only geophysical instrument that can non-invasively estimate water storage changes at field scales comparable to those considered by satellite missions. This property of SGs is a clear advantage over the lysimeters whose installation disturbs the soil profile and provides local measurements of water storage. In the specific case of ET, the proposed technique has an additional advantage in that it does not require any information on the crop type or soil moisture in the radius of influence.

## 5. Conclusions

This study proposed a novel methodology to validate remote sensing estimates of evapotranspiration (ET) using mesoscale ground-truth ET data obtained from a superconducting gravimeter. This validation was consistent since the radius of influence of the superconducting gravimeters (50 to 4000 m) is similar to the horizontal resolution of several satellite observations (e.g., Global Land Evaporation Amsterdam and Priestley–Taylor Jet Propulsion Laboratory Model). In particular, the MOD16A2 products used in this

study have a spatial resolution of 500 m, which provides good representativeness on a global scale.

The gravity-based technique estimates ET as a residual of the classical water balance equation using precipitation and superconducting gravimeter data. The ET gravity-based approach was used to validate the global ET products MOD16A2 in a study site located in Buenos Aires Province for one year from May 2017 to April 2018. This analysis was performed at two different timescales: 8 days and one month. At the 8-day scale, a greater dispersion between the ET time series was observed, with RMSE values of 11.9 mm ($1.5$ mm day$^{-1}$) and a correlation coefficient r = 0.4. The observed discrepancies could be related to the persistence of a periodic 14-day non-hydrological signal in the gravity residuals. This effect had a direct impact on the ET data, resulting in some unrealistic values.

At the monthly timescale, both ET time series showed similar seasonal patterns. The estimated RMSE value for this analysis was 32.6 mm month$^{-1}$ ($1.1$ mm day$^{-1}$) which is within the expected range for satellite products. Furthermore, a better correlation was observed between the ET gravity-based approach and satellite data ($r = 0.7$), as the 14-day periodic signal was attenuated by the temporal upscaling. This result shows the potential of the gravity-based approach as a source of ground-truth ET data on a monthly timescale and spatial mesoscale, which represents a major step forward in the field of remote sensing validation.

The current state of the art for gravimetric data processing can only guarantee good ET estimates at monthly scales. For shorter timescales, highly accurate gravimetric reductions are required to assess ET patterns. However, with the increasing use of superconducting gravimetry, it is expected that data reduction models will improve, leading to better results on smaller timescales.

**Author Contributions:** Conceptualization, R.E.R. and L.G.; methodology, J.P. and M.F.D.; software, J.P.; validation, J.P. and M.F.D.; formal analysis, J.P., M.F.D., L.G. and R.E.R.; investigation, J.P., M.F.D., L.G. and R.E.R.; resources, J.P., M.F.D. and L.G.; data curation, J.P. and M.F.D.; writing—original draft preparation, J.P. and L.G.; writing—review and editing, M.F.D.; visualization, J.P. and M.F.D.; supervision, R.E.R. and L.G.; project administration, J.P., M.F.D., L.G. and R.E.R.; funding acquisition, J.P., M.F.D., L.G. and R.E.R. All authors have read and agreed to the published version of the manuscript.

**Funding:** This research received no external funding.

**Data Availability Statement:** The SG data were published in [43] (https://dataservices.gfz-potsdam.de/panmetaworks/showshort.php?id=escidoc:3748903), accessed on 10 July 2023. The MOD16 Data were obtained from the Google Earth Engine (https://developers.google.com/earth-engine/datasets/catalog/MODIS_006_MOD16A2), accessed on 27 September 2022.

**Acknowledgments:** The authors would like to thank the Instituto de Hidrología de Llanuras "Eduardo Jorge Usunoff", the Consejo Nacional de Investigaciones Científicas Técnicas, the Facultad de Ciencias Astronómicas y Geofísicas, the Facultad de Ciencias Naturales y Museo and the Comisión de Inves-tigaciones Científicas de la provincia de Buenos Aires.

**Conflicts of Interest:** The authors declare no conflict of interest.

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
