# Peer review of "Superconducting Gravimeters: A Novel Tool for Validating Remote Sensing Evapotranspiration Products"

_hydrology, doi:10.3390/hydrology10070146_

Round 1
Reviewer 1 Report
The submitted paper is a scientific paper and the author has explained all the points correctly. The methods are well explained and the paper is methodologically sound. Results are presented according to the objectives of the study. The results are well discussed and they are able to draw a fruitful conclusion. However, the discussion section is missing in this paper which needs to be added as a separate section. In the discussion or conclusion section, it is better to add some limitations that have been observed in this study. Also, add some more recent related works (preferably from the year 2022-2023) as review works in the introduction section.
Author Response
Dear Reviewer,
Thank you for your contributions, they have been of great interest to improve our work.
Best regards,
The authors.
Response to the comments
Note that that the reviewers’ comments are written in bold
Reviewer 1
Comments and Suggestions for Authors
C-R1. - The submitted paper is a scientific paper and the author has explained all the points correctly. The methods are well explained and the paper is methodologically sound. Results are presented according to the objectives of the study. The results are well discussed and they are able to draw a fruitful conclusion.
C-R1a: the discussion section is missing in this paper which needs to be added as a separate section.
R-Authors: We agree with Reviewer 1 (the missing has also been noted by Reviewer 2) and in the revised version of the manuscript we have included a Discussion section.
C-R1b: In the discussion or conclusion section, it is better to add some limitations that have been observed in this study.
R-Authors: The limitations observed in the study and other aspects of the proposed technique are now included in the new Discussion section.
C-R1c: add some more recent related works (preferably from the year 2022-2023) as review works in the introduction section.
R-Authors: We agree, we have include more current references [3, 25-27, 29-31, 54]
Reviewer 2 Report
I have reviewed your manuscript "Superconducting gravimeters: a novel tool for validating remote sensing evapotranspiration products". The subject matter is significant and presents an interesting new approach towards evapotranspiration validation. However, I recommend a major review before this paper can be considered for publication. Below, I have outlined my specific comments and suggestions.
1. Introduction: This section could benefit from a more comprehensive review of the existing literature, particularly on previous studies that have assessed evapotranspiration retrievals from satellite observations. This would establish a solid foundation for your research, allowing readers to understand where your work fits into the broader context. Additionally, providing precise examples of the methods previously employed for the evaluation of remotely sensed evapotranspiration would give readers more clarity.
2. Methods: I noticed that the paper would be strengthened by explaining the methodology in more detail. For example, could you provide the specific steps taken in your methodology, and how exactly these superconducting gravimeters are being used in this context? This would aid in the reproducibility and transparency of your study. In other words, in addition to the technical description provided in paragraph 124-135, a detailed description of the protocol followed to obtain the measurement is required. (Does the method require any calibration?)
3. Section 2.2 (Study Site): This section requires further detail on the study area. It would be advantageous to provide a more detailed description of the percentage of forest and grassland in the study area. This will allow for more nuanced discussion when you interpret your results, as different land cover types can significantly impact evapotranspiration rates.
4. Figure 1: It would be helpful for the readers to include pictures of the study site and the superconducting gravimeters used in your study. This will add depth to your report and make it more engaging and understandable for readers not intimately familiar with these tools.
5. Comparison with previous studies: Please clarify how your method using superconducting gravimeters differs from those of previous studies. Specifically, it would be beneficial to compare your work with studies like those conducted by Carrière et al. (2021) (https://doi.org/10.1029/2021GL096579) and Chaffaut et al. (2020) (https://doi.org/10.1007/1345_2020_105). This comparison would further highlight the novel aspects of your study.
6. Section 2.5 (Validation Metrics): Could you provide more explicit details on the usage of each validation metric in this section? Readers would benefit from understanding why you've selected these specific metrics and how they contribute to the overall evaluation of MODIS product.
7. Benchmark Metrics: Does the field have a benchmark or target metric similar to the SMAP's mission target of ubRMSD = 0.04 for evaluating soil moisture retrievals? If such a benchmark exists, how do your results compare to this standard? If no such benchmark exists, providing a rationale or suggestion for an appropriate benchmark would add depth to the discussion.
8. Discussion Section: A discussion of the obtained results seems to be missing. This section is crucial in interpreting the data and drawing meaningful conclusions. The readers would benefit from a comprehensive discussion of the findings, including their implications, limitations, and potential for future work.
9. Line 358: In this line, please note that the SMAP satellite has a relatively coarse resolution of approximately 33 km.
I hope these comments will aid you in improving the manuscript.
Can be improved.
Author Response
Dear Reviewer,
Thank you for your contributions, they have been of great interest to improve our work.
Best regards,
The authors.
Response to the comments
Note that that the reviewers’ comments are written in bold
Reviewer 2
Comments and Suggestions for Authors
C-R2. - Introduction: This section could benefit from a more comprehensive review of the existing literature, particularly on previous studies that have assessed evapotranspiration retrievals from satellite observations. This would establish a solid foundation for your research, allowing readers to understand where your work fits into the broader context. Additionally, providing precise examples of the methods previously employed for the evaluation of remotely sensed evapotranspiration would give readers more clarity.
R-Authors: Your observation is correct. We have included the information between lines 90-93, as well as the respective references [24-30].
C-R2. - Methods: I noticed that the paper would be strengthened by explaining the methodology in more detail. For example, could you provide the specific steps taken in your methodology, and how exactly these superconducting gravimeters are being used in this context? This would aid in the reproducibility and transparency of your study. In other words, in addition to the technical description provided in paragraph 124-135, a detailed description of the protocol followed to obtain the measurement is required. (Does the method require any calibration?)
R-Authors:We agree with this comment and we include more details in the description of the methodology. At the end of Section 2.4 we include a brief summary of the steps necessary to obtain ET starting from the raw SG data.
C-R2. - Section 2.2 (Study Site): This section requires further detail on the study area. It would be advantageous to provide a more detailed description of the percentage of forest and grassland in the study area. This will allow for more nuanced discussion when you interpret your results, as different land cover types can significantly impact evapotranspiration rates.
R-Authors: We thank the reviewer's suggestion and the following sentences are added:
The AGGO site is covered by natural grass and surrounded by a eucalyptus forest. The climate is temperate, with a mean annual temperature, relative humidity, and precipitation 15.8°C, 74.3%, and 1007 mm, respectively the soil is composed of silty and clayey sediments and the mean groundwater level is approximately 13 m depth.
C-R2. - Figure 1: It would be helpful for the readers to include pictures of the study site and the superconducting gravimeters used in your study. This will add depth to your report and make it more engaging and understandable for readers not intimately familiar with these tools.
R-Authors: We agree with the Reviewer and we include pictures of the study site and SG in Figure 1.
C-R2. - Comparison with previous studies: Please clarify how your method using superconducting gravimeters differs from those of previous studies. Specifically, it would be beneficial to compare your work with studies like those conducted by Carrière et al. (2021) (https://doi.org/10.1029/2021GL096579) and Chaffaut et al. (2020) (https://doi.org/10.1007/1345_2020_105). This comparison would further highlight the novel aspects of your study.
R-Authors: We thank the Reviewer for this comment. In the revised version of the manuscript we added a brief description of three papers [2, 21-22] that apply different methodologies to estimate ET using superconducting gravimetric data in the introduction section (line 79).
The comparison between Chaffaut et al. (2020) and our work is not included, as they do not estimate evapotranspiration. Chaffaut et al. estimate water storage changes using data from superconducting gravimeters in a mountainous area and compare these estimates with those provided by two other independent methods.
C-R2. - Section 2.5 (Validation Metrics): Could you provide more explicit details on the usage of each validation metric in this section? Readers would benefit from understanding why you've selected these specific metrics and how they contribute to the overall evaluation of MODIS product.
R-Authors: We thank the Reviewer for the comment and add the following paragraph at the end of Section 2.5:
“RMSE and MAE are two of the most commonly used metrics in model evaluation in the physical and environmental sciences [53-54]. These statistical metrics are easy to interpret, as lower values indicate better model performance. One of the main differences between these metrics is that the RMSE is more sensitive to outliers and extreme values. However, both metrics should be used together since there is no consensus on which metric is the most appropriate for understanding the agreement between ET estimates. Pearson correlation coefficients r quantify the linear correlation between ETM and ET gravity-based approach.”
C-R2. - Benchmark Metrics: Does the field have a benchmark or target metric similar to the SMAP's mission target of ubRMSD = 0.04 for evaluating soil moisture retrievals? If such a benchmark exists, how do your results compare to this standard? If no such benchmark exists, providing a rationale or suggestion for an appropriate benchmark would add depth to the discussion.
R-Authors: In order to clarify this point we add the following sentences in the new Discussion Section:
The proposed method does not require any constraints related to soil moisture to estimate ET. The superconducting gravimeter implicitly records the variations of soil moisture, so it is not necessary to measure or model the water content in the soil profile.
C-R2. - Discussion Section: A discussion of the obtained results seems to be missing. This section is crucial in interpreting the data and drawing meaningful conclusions. The readers would benefit from a comprehensive discussion of the findings, including their implications, limitations, and potential for future work.
R-Authors: The missing of the Discussion section has also been pointed out by Reviewer 1. In the new version of the manuscript we include a new section (4) with the discussion of the results and different aspects of the proposed technique. We also moved to this section some sentences that were originally included in the Results section.
C-R2. - Line 358: In this line, please note that the SMAP satellite has a relatively coarse resolution of approximately 33 km.
R-Authors: We agree with the Reviewer and remove the reference of SMAP in line 358.
Reviewer 3 Report
Well written and very interesting. I note you avoid associating a specific spatial support to the resulting measurements, which some readers are going to find odd ... having an estimated quantity as a depth of water, but then not knowing just what area to which that applies ...
Line 31 - You do not need an indefinite article to start this sentence. Just start with "Evapotranspiration (ET) is one ..."
Line 46 - The comma in this line should not be there.
Line 47 - Replace "estimate" with "estimation"
Line 54 - This should be "However, lysimeter data describe ET at a single point in ..."
Line 85 - This should be "Nevertheless, there are few gorund stations available to estimate ..."
Line 87 - Replace "to monitor" with "for monitoring"
Line 103 - " ... which has continuously measured temporal gravity changes since ..."
Line 124 - Replace "to monitor" with "for monitoring"
Line 137 - Replace "Then" with "Thus"
Line 163 - Replace "radius" with "radii"
Line 175 - Start this sentence with "An"
Author Response
Dear Reviewer,
Thank you for your contributions, they have been of great interest to improve our work.
Best regards,
The authors.
Response to the comments
Note that that the reviewers’ comments are written in bold
Reviewer 3
C-R3. - Well written and very interesting. I note you avoid associating a specific spatial support to the resulting measurements, which some readers are going to find odd ... having an estimated quantity as a depth of water, but then not knowing just what area to which that applies ...
R-Authors: We thank the comment made by the Reviewer. We add a new discussion Section in the revised version of the manuscript as suggested by Reviewers 1 and 2. In this Section we mention that the radius of influence of the SG depends on the depth of the water table in the study area. Furthermore, the SG radius of influence at AGGO estimated from Eqn. (2) is 1300 m.
C-R3. - Line 31 - You do not need an indefinite article to start this sentence. Just start with "Evapotranspiration (ET) is one ..."
R-Authors: We replace “The evapotranspiration” by “Evapotranspiration”
C-R3. - Line 46 - The comma in this line should not be there.
R-Authors: We remove the comma
C-R3. - Line 47 - Replace "estimate" with "estimation"
R-Authors: We replace “estimate” by “estimation”
C-R3. - Line 54 - This should be "However, lysimeter data describe ET at a single point in ..."
R-Authors: We replace “However, lysimeter data describe the ET at a single point in” by “However, lysimeter data describe ET at a single point in”
C-R3. - Line 85 - This should be "Nevertheless, there are few gorund stations available to estimate ..."
R-Authors: We replace “Nevertheless, there are few ground stations to estimate ET” by “Nevertheless, there are few ground stations available to estimate ET”
C-R3. - Line 87 - Replace "to monitor" with "for monitoring"
R-Authors: We replace “to monitor” by “for monitoring”
C-R3. - Line 103 - " ... which has continuously measured temporal gravity changes since ..."
R-Authors. - We replace “which continuously measures the temporal gravity changes since” by “which has continuously measured temporal gravity changes since”
C-R3. - Line 124 - Replace "to monitor" with "for monitoring"
R-Authors: We replace “to monitor” by “for monitoring”
C-R3. - Line 137 - Replace "Then" with "Thus"
R-Authors: We replace “Then” by “Thus”
C-R3. - Line 163 - Replace "radius" with "radii"
R-Authors: We replace “radius” by “radii”
C-R3. - Line 175 - Start this sentence with "An"
R-Authors: We add the word “An” at the beginning of the sentence
Round 2
Reviewer 2 Report
Firstly, I would like to extend my gratitude to the authors for their detailed responses to my previous comments. The manuscript has improved considerably and offers a significant contribution to the field. I recommend the acceptance of the paper after minor revisions as outlined below:
Line 31:
Since the abbreviation for evapotranspiration (ET) has already been introduced in the abstract, there's no need to repeat it in the main text. I suggest starting the paragraph as: "Evapotranspiration is one of the major..."
Line 34-35:
The statement ending with "developing effective water management strategies" requires citations to provide evidence for the claim. For example, the following studies could be referenced to illustrate ET's significance on regional and global scales:
- "Integrating Satellite Imagery and Ground-Based Measurements with a Machine Learning Model for Monitoring Lake Dynamics over a Semi-Arid Region" (https://doi.org/10.3390/hydrology10040078)
- "A Global Assessment of Terrestrial Evapotranspiration Increase Due to Surface Water Area Change" (https://doi.org/10.1029/2018EF001066)
- "Evaporative water loss of 1.42 million global lakes" (https://www.nature.com/articles/s41467-022-31125-6)
Lines 42-44:
The statement regarding the monitoring of individual water storage components could be bolstered with references to relevant studies. These studies demonstrate the practices of monitoring components such as streamflow, soil moisture, and snowpack individually (using remote sensing and in-situ observations:
- Snowpack: "A comparison of National Water Model retrospective analysis snow outputs at snow telemetry sites across the Western United States" (https://doi.org/10.1002/hyp.14469)
- Streamflow: "Assessing the National Water Model’s Streamflow Estimates Using a Multi-Decade Retrospective Dataset across the Contiguous United States" (https://doi.org/10.3390/w15132319)
- Soil moisture: "Assessing the Spatiotemporal Variability of SMAP Soil Moisture Accuracy in a Deciduous Forest Region" (https://doi.org/10.3390/rs14143329)
Figure 1:
I appreciate the details provided in Figure 1. It significantly aids in understanding the presented concepts and measurement application.
Line 234:
Please provide a direct link to the database referenced here. It could be added as a citation for easy access to the readers.
Section 5. Patents:
The manuscript does not make it clear whether the work includes any patent. This information should be explicitly stated for clarity and transparency.
In conclusion, this study provides a valuable exploration of the application of superconducting gravimeters for validating remote sensing evapotranspiration products. The above minor revisions will ensure the manuscript's clarity and thoroughness.
Moderate editing of the English language required
Author Response
Dear rewiever,
The authors would like to express their gratitude for your valuable comments and suggestions, as they have played a pivotal role in enhancing the quality of the manuscript.
Sincerely,
The authors.
Response to the comments
Note that that the reviewer’ comments are written in bold
Reviewer 2
Comments and Suggestions for Authors
Firstly, I would like to extend my gratitude to the authors for their detailed responses to my previous comments. The manuscript has improved considerably and offers a significant contribution to the field. I recommend the acceptance of the paper after minor revisions as outlined below:
Line 31:
Since the abbreviation for evapotranspiration (ET) has already been introduced in the abstract, there's no need to repeat it in the main text. I suggest starting the paragraph as: "Evapotranspiration is one of the major..."
R-Authors: Thank you for the observation, we have removed the abbreviation in the introduction.
Line 34-35:
The statement ending with "developing effective water management strategies" requires citations to provide evidence for the claim. For example, the following studies could be referenced to illustrate ET's significance on regional and global scales:
- "Integrating Satellite Imagery and Ground-Based Measurements with a Machine Learning Model for Monitoring Lake Dynamics over a Semi-Arid Region" (https://doi.org/10.3390/hydrology10040078)
- "A Global Assessment of Terrestrial Evapotranspiration Increase Due to Surface Water Area Change" (https://doi.org/10.1029/2018EF001066 )
- "Evaporative water loss of 1.42 million global lakes" (https://www.nature.com/articles/s41467-022-31125-6)
R-Authors: We agree, we have included references [1-3].
Lines 42-44:
The statement regarding the monitoring of individual water storage components could be bolstered with references to relevant studies. These studies demonstrate the practices of monitoring components such as streamflow, soil moisture, and snowpack individually (using remote sensing and in-situ observations:
- Snowpack: "A comparison of National Water Model retrospective analysis snow outputs at snow telemetry sites across the Western United States" (https://doi.org/10.1002/hyp.14469)
- Streamflow: "Assessing the National Water Model’s Streamflow Estimates Using a Multi-Decade Retrospective Dataset across the Contiguous United States" (https://doi.org/10.3390/w15132319)
- Soil moisture: "Assessing the Spatiotemporal Variability of SMAP Soil Moisture Accuracy in a Deciduous Forest Region" (https://doi.org/10.3390/rs14143329)
R-Authors: We agree, we have included references [5-7].
Figure 1:
I appreciate the details provided in Figure 1. It significantly aids in understanding the presented concepts and measurement application.
R-Authors: Thank you for yours comments
Line 234:
Please provide a direct link to the database referenced here. It could be added as a citation for easy access to the readers.
R-Authors: The database (gravity residuals and precipitation time series) is cited as Mikolaj et al. (2018) [43] and the link is https://doi.org/10.5880/GFZ.5.4.2018.001. Moreover, we provide a direct link to the database in Data Availability Statement Sections.
Section 5. Patents:
The manuscript does not make it clear whether the work includes any patent. This information should be explicitly stated for clarity and transparency.
R-Authors: The manuscript does not include patents, we have decided not to indicate anything about it, according to what is stated in the template "Patents: This section is not mandatory but may be added if there are patents resulting from the work reported in this manuscript”
In conclusion, this study provides a valuable exploration of the application of superconducting gravimeters for validating remote sensing evapotranspiration products. The above minor revisions will ensure the manuscript's clarity and thoroughness.
In addition, we did a thorough language review. In the attach file, we have highlighted in yellow the changes made.
Round 3
Reviewer 2 Report
I recommend the acceptance in the present form.